# Learning convolution filters for inverse covariance estimation of neural network connectivity

**George O. Mohler**[*]
Department of Mathematics and Computer Science
Santa Clara University University
Santa Clara, CA, USA
gmohler@scu.edu

## Abstract

We consider the problem of inferring direct neural network connections from Calcium imaging time series. Inverse covariance estimation has proven to be a fast and accurate method for learning macro- and micro-scale network connectivity in the brain and in a recent Kaggle Connectomics competition inverse covariance was the main component of several top ten solutions, including our own and the winning team's algorithm. However, the accuracy of inverse covariance estimation is highly sensitive to signal preprocessing of the Calcium fluorescence time series. Furthermore, brute force optimization methods such as grid search and coordinate ascent over signal processing parameters is a time intensive process, where learning may take several days and parameters that optimize one network may not generalize to networks with different size and parameters. In this paper we show how inverse covariance estimation can be dramatically improved using a simple convolution filter prior to applying sample covariance. Furthermore, these signal processing parameters can be learned quickly using a supervised optimization algorithm. In particular, we maximize a binomial log-likelihood loss function with respect to a convolution filter of the time series and the inverse covariance regularization parameter. Our proposed algorithm is relatively fast on networks the size of those in the competition (1000 neurons), producing AUC scores with similar accuracy to the winning solution in training time under 2 hours on a cpu. Prediction on new networks of the same size is carried out in less than 15 minutes, the time it takes to read in the data and write out the solution.

## 1 Introduction

Determining the topology of macro-scale functional networks in the brain and micro-scale neural networks has important applications to disease diagnosis and is an important step in understanding brain function in general [11, 19]. Modern neuroimaging techniques allow for the activity of hundreds of thousands of neurons to be simultaneously monitored [19] and recent algorithmic research has focused on the inference of network connectivity from such neural imaging data. A number of approaches to solve this problem have been proposed, including Granger causality [3], Bayesian networks [6], generalized transfer entropy [19], partial coherence [5], and approaches that directly model network dynamics [16, 18, 14, 22].

---
[*]

Several challenges must be overcome when reconstructing network connectivity from imaging data. First, imaging data is noisy and low resolution. The rate of neuron firing may be faster than the image sampling rate [19] and light scattering effects [13, 19] lead to signal correlations at short distances irrespective of network connectivity. Second, causality must be inferred from observed correlations in neural activity. Neuron spiking is highly correlated both with directly connected neurons and those connected through intermediate neurons. Coupled with the low sampling rate this poses a significant challenge, as it may be the case that neuron $i$ triggers neuron $j$, which then triggers neuron $k$, all within a time frame less than the sampling rate.

To solve the second challenge, sparse inverse covariance estimation has recently become a popular technique for disentangling causation from correlation [11, 15, 23, 1, 9, 10]. While the sample covariance matrix only provides information on variable correlations, zeros in the inverse covariance matrix correspond to conditional independence of variables under normality assumptions on the data. In the context of inferring network connectivity from leaky integrate and fire neural network time-series, however, it is not clear what set of random variables one should use to compute sample covariance (a necessary step for estimating inverse covariance). While the simplest choice is the raw time-series signal, the presence of both Gaussian and jump-type noise make this significantly less accurate than applying signal preprocessing aimed at filtering times at which neurons fire.

In a recent Kaggle competition focused on inferring neural network connectivity from Calcium imaging time series, our approach used inverse covariance estimation to predict network connections. Instead of using the raw time series to compute sample covariance, we observed improved Area Under the Curve (receiver operating characteristic [2]) scores by thresholding the time derivative of the time-series signal and then combining inverse covariance corresponding to several thresholds and time-lags in an ensemble. This is similar to the approach of the winning solution [21], though they considered a significantly larger set of thresholds and nonlinear filters learned via coordinate ascent, the result of which produced a private leaderboard AUC score of .9416 compared to our score of .9338. However, both of these approaches are computationally intensive, where prediction on a new network alone takes 10 hours in the case of the winning solution [21]. Furthermore, parameters for signal processing were highly tuned for optimizing AUC of the competition networks and don't generalize to networks of different size or parameters [21]. Given that coordinate ascent takes days for learning parameters of new networks, this makes such an approach impractical.

In this paper we show how inverse covariance estimation can be significantly improved by applying a simple convolution filter to the raw time series signal. The filter can be learned quickly in a supervised manner, requiring no time intensive grid search or coordinate ascent. In particular, we optimize a smooth binomial log-likelihood loss function with respect to a time series convolution kernel, along with the inverse covariance regularization parameter, using L-BFGS [17]. Training the model is fast and accurate, running in under 2 hours on a CPU and producing AUC scores that are competitive with the winning Kaggle solution. The outline of the paper is as follows. In Section 2 we review inverse covariance estimation and introduce our convolution based method for signal preprocessing. In Section 3 we provide the details of our supervised learning algorithm and in Section 4 we present results of the algorithm applied to the Kaggle Connectomics dataset.

## 2 Modeling framework for inferring neural connectivity

### 2.1 Background on inverse covariance estimation

Let $\mathbf{X} \in \mathbb{R}^{n \times p}$ be a data set of $n$ observations from a multivariate Gaussian distribution with $p$ variables, let $\boldsymbol{\Sigma}$ denote the covariance matrix of the random variables, and $\mathbf{S}$ the sample covariance. Variables $i$ and $j$ are conditionally independent given all other variables if the $ij$th component of $\boldsymbol{\Theta} = \boldsymbol{\Sigma}^{-1}$ is zero. For this reason, a popular approach for inferring connectivity in sparse networks is to estimate the inverse covariance matrix via $l_1$ penalized maximum likelihood,

$$\hat{\boldsymbol{\Theta}} = \arg\max_{\boldsymbol{\Theta}} \left\{ \log\Big( \det(\boldsymbol{\Theta}) \Big) - tr(\mathbf{S}\boldsymbol{\Theta}) - \lambda\|\boldsymbol{\Theta}\|_1 \right\}, \tag{1}$$

[11, 15, 23, 1, 9, 10], commonly referred to as GLASSO (graphical least absolute shrinkage and selection operator). GLASSO has been used to infer brain connectivity for the purpose of diagnosing Alzheimer's disease [11] and determining brain architecture and pathologies [23].

While GLASSO is a useful method for imposing sparsity on network connections, in the Kaggle Connectomics competition AUC was the metric used for evaluating competing models and on AUC GLASSO only performs marginally better (AUC$\approx$ .89) than the generalized transfer entropy Kaggle benchmark (AUC$\approx$ .88). The reason for the poor performance of GLASSO on AUC is that $l_1$ penalization forces a large percentage of neuron connection scores to zero, whereas high AUC performance requires ranking all possible connections.

We therefore use $l_2$ penalized inverse covariance estimation [23, 12],

$$\hat{\boldsymbol{\Theta}} = \left(\mathbf{S} + \lambda\mathbf{I}\right)^{-1}, \tag{2}$$

instead of optimizing Equation 1. While one advantage of Equation 2 is that all connections are assigned a non-zero score, another benefit is derivatives with respect to model parameters are easy to determine and compute using the standard formula for the derivative of an inverse matrix. In particular, our model consists of parametrizing $\mathbf{S}$ using a convolution filter applied to the raw Calcium fluorescence time series and Equation 2 facilitates derivative based optimization. We return to GLASSO in the discussion section at the end of the paper.

## 2.2 Signal processing

Next we introduce a model for the covariance matrix $\mathbf{S}$ taking as input observed imaging data from a neural network. Let $\mathbf{f}$ be the Calcium fluorescence time series signal, where $f_t^i$ is the signal observed at neuron $i$ in the network at time $t$. The goal in this paper is to infer direct network connections from the observed fluorescence time series (see Figure 1). While $f_t^i$ can be used directly to calculate

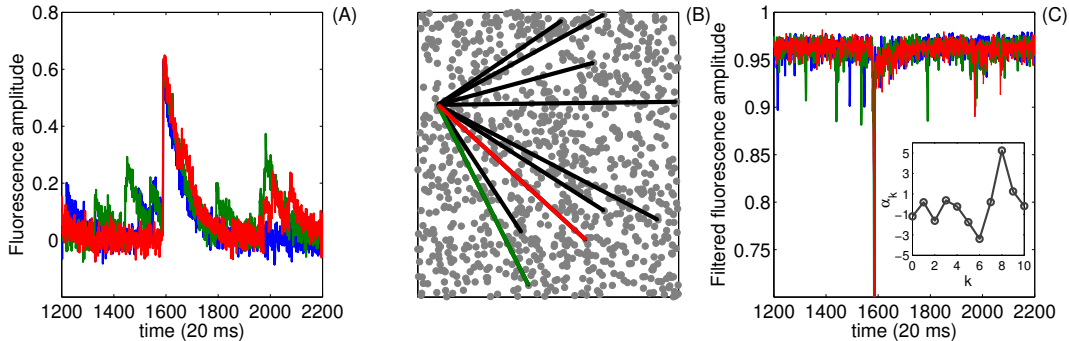

Figure 1: (A) Fluorescence time series $\mathbf{f}^i$ for neuron $i = 1$ (blue) of Kaggle Connectomics network 2 and time series for two neurons (red and green) connected to neuron 1. Synchronized firing of all 1000 neurons occurs around time 1600. (B) Neuron locations (gray) in network 2 and direct connections to neuron 1 (green and red connections correspond to time series in Fig 1A). The task is to reconstruct network connectivity as in Fig 1B for all neurons given time series data as in Fig 1A. (C) Filtered fluorescence time series $\sigma(\mathbf{f}^i * \boldsymbol{\alpha} + \alpha^{\text{bias}})$ using the convolution kernel $\boldsymbol{\alpha}$ (inset figure) learned from our method detailed in Section 3.

covariance between fluorescence time series, significant improvements in model performance are achieved by filtering the signal to obtain an estimate of $n_t^i$, the number of times neuron $i$ fired between $t$ and $t + \Delta t$. In the competition we used simple thresholding of the time series derivative

$\Delta f_t^i = f_{t+\Delta t}^i - f_t^i$ to estimate neuron firing times,

$$n_t^i = 1_{\{\Delta f_t^i > \mu\}}. \tag{3}$$

The covariance matrix was then computed using a variety of threshold values $\mu$ and time-lags $k$. In particular, the $(i,j)$th entry of $\mathbf{S}(\mu, k)$ was determined by,

$$s_{ij} = \frac{1}{T} \sum_{t=k}^{T} (n_t^i - \overline{n}^i)(n_{t-k}^j - \overline{n}^j), \tag{4}$$

where $\overline{n}^i$ is the mean signal. The covariance matrices were then inverted using Equation 2 and combined using LambdaMart [4] to optimize AUC, along with a restricted Boltzmann machine and generalized linear model. In Figure 2, we illustrate the sensitivity of inverse covariance estimation on the threshold parameter $\mu$, regularization parameter $\lambda$, and time-lag parameter $k$. Using the raw time series signal leads to AUC scores between 0.84 and 0.88, whereas for good choices of the threshold and regularization parameter Equation 2 yields AUC scores above 0.92. Further gains are achieved by using an ensemble over varying $\mu$, $\lambda$, and $k$.

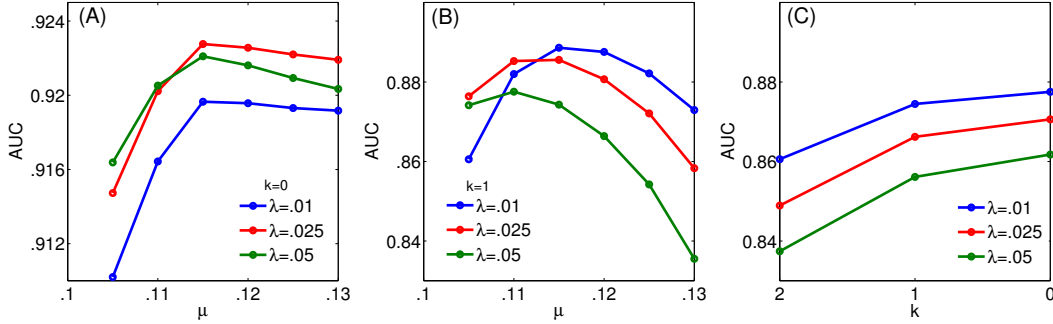

Figure 2: (A) AUC scores for network 2 using Equations 2, 3, and 4 with a time lag of $k = 0$ and varying threshold $\mu$ and regularization parameter $\lambda$. (B) AUC scores analogous to Figure 2A, but for a time lag of $k = 1$. (C) AUC scores corresponding to inverse covariance estimation using raw time series signal. For comparison, generalized transfer entropy [19] corresponds to AUC$\approx .88$ and simple correlation corresponds to AUC$\approx .66$.

In this paper we take a different approach in order to jointly learn the processed fluorescence signal and the inverse covariance estimate. In particular, we convolve the fluorescence time series $\mathbf{f}^i$ with a kernel $\boldsymbol{\alpha}$ and then pass the convolution through the logistic function $\sigma(x)$,

$$\mathbf{y}^i = \sigma(\mathbf{f}^i * \boldsymbol{\alpha} + \alpha^{\text{bias}}). \tag{5}$$

Note for $\alpha_0 = -\alpha_1$ (and $\alpha_k = 0$ otherwise) this convolution filter approximates the threshold filter in Equation 3. However, it turns out that the learned optimal filter is significantly different than time derivative thresholding (see Figure 1C). Inverse covariance is then estimated via Equation 2, where the sample covariance is given by,

$$s_{ij} = \frac{1}{T} \sum_{t=1}^{T} (y_t^i - \overline{y}^i)(y_t^j - \overline{y}^j). \tag{6}$$

The time lags no longer appear in Equation 6, but instead are reflected in the convolution filter.

## 2.3 Supervised inverse covariance estimation

Given the sensitivity of model performance on signal processing illustrated in Figure 2, our goal is now to learn the optimal filter $\alpha$ by optimizing a smooth loss function. To do this we introduce a model for the probability of neurons being connected as a function of inverse covariance.

Let $z_{ij} = 1$ if neuron $i$ connects to neuron $j$ and zero otherwise and let $\Theta(\alpha, \lambda)$ be the inverse covariance matrix that depends on the smoothing parameter $\lambda$ from Section 2.1 and the convolution filter $\alpha$ from Section 2.2. We model the probability of neuron $i$ connecting to $j$ as $\sigma_{ij} = \sigma(\theta_{ij}\beta_0 + \beta_1)$ where $\sigma$ is the logistic function and $\theta_{ij}$ is the $(i,j)$th entry of $\Theta$. In summary, our model for scoring the connection from $i$ to $j$ is detailed in Algorithm 1.

---
**Algorithm 1:** Inverse covariance scoring algorithm

---
**Input**: $\mathbf{f}\ \alpha\ \alpha^{\text{bias}}\ \lambda\ \beta_0\ \beta_1$ $\quad$ \\ fluorescence signal and model parameters
$\mathbf{y}^i = \sigma(\mathbf{f}^i * \alpha + \alpha^{\text{bias}})$ $\quad$ \\ apply convolution filter and logistic function to signal
**for** $i \leftarrow 1$ **to** $N$ **do**
$\quad$ **for** $j \leftarrow 1$ **to** $N$ **do**
$\quad\quad$ $s_{ij} = \frac{1}{T}\sum_{t=1}^{T}(y_t^i - \overline{y}^i)(y_t^j - \overline{y}^j)$ $\quad$ \\ compute sample covariance matrix
$\quad$ **end**
**end**
$\Theta = (\mathbf{S} + \lambda\mathbf{I})^{-1}$ $\quad$ \\ compute inverse covariance matrix
**Output**: $\sigma(\Theta\beta_0 + \beta_1)$ $\quad$ \\ output connection probability matrix

---

The loss function we aim to optimize is the binomial log-likelihood, given by,

$$L(\alpha, \lambda, \beta_0, \beta_1) = \sum_{i \neq j} \chi z_{ij}\log(\sigma_{ij}) + (1-\chi)(1-z_{ij})\log(1-\sigma_{ij}), \tag{7}$$

where the parameter $\chi$ is chosen to balance the dataset. The networks in the Kaggle dataset are sparse, with approximately 1.2% connections, so we choose $\chi = .988$. For $\chi$ values within 10% of the true percentage of connections, AUC scores are above .935. Without data balancing, the model achieves an AUC score of .925, so the introduction of $\chi$ is important. While smooth approximations of AUC are possible, we find that optimizing Equation 7 instead still yields high AUC scores.

To use derivative based optimization methods that converge quickly, we need to calculate the derivatives of Equation 7. Defining,

$$\omega_{ij} = \chi z_{ij}(1 - \sigma_{ij}) - (1-\chi)(1-z_{ij})\sigma_{ij}, \tag{8}$$

then the derivatives of the loss function with respect to the model parameters are specified by,

$$\frac{dL}{d\beta_0} = \sum_{i \neq j}\omega_{ij}\theta_{ij}, \quad \frac{dL}{d\beta_1} = \sum_{i \neq j}\omega_{ij}, \tag{9}$$

$$\frac{dL}{d\lambda} = \sum_{i \neq j}\beta_0\omega_{ij}\frac{d\theta_{ij}}{d\lambda}, \quad \frac{dL}{d\alpha_k} = \sum_{i \neq j}\beta_0\omega_{ij}\frac{d\theta_{ij}}{d\alpha_k}. \tag{10}$$

Using the inverse derivative formula, we have that the derivatives of the inverse covariance matrix satisfy the following convenient equations,

$$\frac{d\Theta}{d\lambda} = -\left((\mathbf{S}(\alpha) + \lambda\mathbf{I})^{-1}\right)^2, \quad \frac{d\Theta}{d\alpha_k} = -(\mathbf{S}(\alpha) + \lambda\mathbf{I})^{-1}\frac{d\mathbf{S}}{d\alpha_k}(\mathbf{S}(\alpha) + \lambda\mathbf{I})^{-1}, \tag{11}$$

where $\mathbf{S}$ is the sample covariance matrix from Section 2.2. The derivatives of the sample covariance are then found by substituting $\frac{dy_t^i}{d\alpha_k} = y_t^i(1 - y_t^i)f_{t-k}^i$ into Equation 6 and using the product rule.

## 3 Results

We test our methodology using data provided through the Kaggle Connectomics competition. In the Kaggle competition, neural activity was modeled using a leaky integrate and fire model outlined in [19]. Four 1000 neuron networks with 179,500 time series observations per network were provided for training, a test network of the same size and parameters was provided without labels to determine the public leaderboard, and final standings were computed using a 6th network for validation. The goal of the competition was to infer the network connections from the observed Fluorescence time series signal (see Figure 1) and the error metric for determining model performance was AUC.

There are two ways in which we determined the size of the convolution filter. The first is through inspecting the decay of cross-correlation as a function of the time-lag. For the networks we consider in the paper, this decay takes place over 10-15 time units. The second method is to add an additional time unit one at a time until cross-validated AUC scores no longer improve. This happens for the networks we consider at 10 time units. We therefore consider a convolution filter with $k = 0...10$.

We use the off-the-shelf optimization method L-BFGS [17] to optimize Equation 7. Prior to applying the convolution filter, we attempt to remove light scattering effects simulated in the competition by inverting the equation,

$$F_t^i = f_t^i + A_{sc} \sum_{j \neq i} f_t^j \exp \left\{ - (d_{ij}/\lambda_{sc})^2 \right\}. \tag{12}$$

Here $F_t^i$ is the observed fluorescence provided for the competition with light scattering effects (see [19]) and $d_{ij}$ is the distance between neuron $i$ and $j$. The parameter values $A_{sc} = .15$ and $\lambda_{sc} = .025$ were determined such that the correlation between neuron distance and signal covariance was approximately zero.

We learn the model parameters using network 2 and training time takes less than 2 hours in Matlab on a laptop with a 2.3 GHz Intel Core i7 processor and 16GB of RAM. Whereas prediction alone takes 10 hours on one network for the winning Kaggle entry [21], prediction using Algorithm 1 takes 15 minutes total and the algorithm itself runs in 20 seconds (the rest of the time is dedicated to reading the competition csv files into and out of Matlab). In Figure 3 we display results for all four of the training networks using 80 iterations of L-BFGS (we used four outer iterations with maxIter= 20 and TolX= $1e - 5$). The convolution filter is initialized to random values and at every 20 iterations we plot the corresponding filtered signal for neuron 1 of network 2 over the first 1000 time series observations. After 10 iterations all four networks have an AUC score above 0.9. After 80 iterations the AUC private leaderboard score of the winning solution is within the range of the AUC scores of networks 1, 3, and 4 (trained on network 2). We note that during training intermediate AUC scores do not increase monotonically and also exhibit several plateaus. This is likely due to the fact that AUC is a non-smooth loss function and we used the binomial likelihood in its stead.

## 4 Discussion

We introduced a model for inferring connectivity in neural networks along with a fast and easy to implement optimization strategy. In this paper we focused on the application to leaky integrate and fire models of neural activity, but our methodology may find application to other types of cross-exciting point processes such as models of credit risk contagion [7] or contagion processes on social networks [20].

It is worth noting that we used a Gaussian model for inverse covariance even though the data was highly non-Gaussian. In particular, neural firing time series data is generated by a nonlinear, mutually-exciting point process. We believe that it is the fact that the input data is non-Gaussian that the signal processing is so crucial. In this case $f_t^i$ and $f_s^j$ are highly dependent for $10 > t - s > 0$

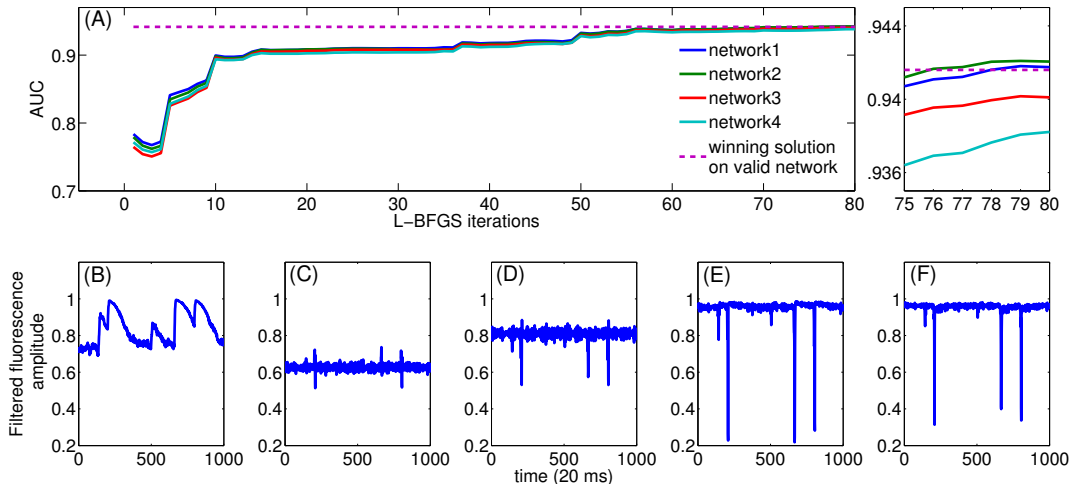

Figure 3: (A) Networks 1-4 AUC values plotted against L-BFGS iterations where network 2 was used to learn the convolution filter. The non-monotonic increase can be attributed to optimizing the binomial log-likelihood rather than AUC directly. (B-F) Every 20 iterations we also plot a subsection of the filtered signal of neuron 1 from network 2. The filter is initially given random values but quickly produces impulse-like signals with high AUC scores. The AUC score of the winning solution is within the range of the AUC scores of held-out networks 1, 3, 4 after 80 iterations of L-BFGS.

and $j \to i$. Empirically, the learned convolution filter compensates for the model mis-specification and allows for the "wrong" model to still achieve a high degree of accuracy.

We also note that using directed network estimation did not improve our methods, nor the methods of other top solutions in the competition. This may be due to the fact that the resolution of Calcium fluorescence imaging is coarser than the timescale of network dynamics, so that directionality information is lost in the imaging process. That being said, it is possible to adapt our method for estimation of directed networks. This can be accomplished by introducing two different filters $\alpha_i$ and $\alpha_j$ into Equations 5 and 6 to allow for an asymmetric covariance matrix $S$ in Equation 6. It would be interesting to assess the performance of such a method on networks with higher resolution imaging in future research.

While the focus here was on AUC maximization, other loss functions may be useful to consider. For sparse networks where the average network degree is known, precision or discounted cumulative gain may be reasonable alternatives to AUC. Here it is worth noting that $l_1$ penalization is more accurate for these types of loss functions that favor sparse solutions. In Table 1 we compare the accuracy of Equation 1 vs Equation 2 on both AUC and PREC@$k$ (where $k$ is chosen to be the known number of network connections). For signal processing we return to time-derivative thresholding and use the parameters that yielded the best single inverse covariance estimate during the competition. While $l_2$ penalization is significantly more accurate for AUC, this is not the case for PREC@$k$ for which GLASSO achieves a higher precision.

It is clear that the sample covariance $\mathbf{S}$ in Equation 1 can be parameterized by a convolution kernel $\boldsymbol{\alpha}$, but supervised learning is no longer as straightforward. Coordinate ascent can be used, but given that Equation 1 is orders of magnitude slower to solve than Equation 2, such an approach may not be practical. Letting $G(\boldsymbol{\Theta}, \mathbf{S})$ be the penalized log-likelihood corresponding to GLASSO in Equation 1, another possibility is to jointly optimize

$$\rho G(\boldsymbol{\Theta}, \mathbf{S}) + (1 - \rho) L(\boldsymbol{\Theta}, \mathbf{S}) \tag{13}$$

|  | $\lambda_{l_1} = 5 \cdot 10^{-5}$ | $\lambda_{l_1} = 1 \cdot 10^{-4}$ | $\lambda_{l_1} = 5 \cdot 10^{-4}$ | $\lambda_{l_2} = 2 \cdot 10^{-2}$ |
|---|---|---|---|---|
| Network1 | .894/**.423** | .884/.420 | .882/.420 | **.926**/.394 |
| Network2 | .894/**.417** | .885/.416 | .885/.415 | **.924**/.385 |
| Network3 | .894/.423 | .885/.425 | .884/**.427** | **.925**/.397 |

Table 1: AUC/PREC@k for $l_1$ vs. $l_2$ penalized inverse covariance estimation (where k equals the true number of connections). Time series preprocessed by a derivative threshold of .125 and removing spikes when 800 or more neurons fire simultaneously. For $l_1$ penalization AUC increases as $\lambda_{l_1}$ decreases, though the Rglasso solver [8] becomes prohibitively slow for $\lambda_{l_1}$ on the order of $10^{-5}$ or smaller.

where $L$ is the binomial log-likelihood in Equation 7. In this case both the convolution filter and the inverse covariance estimate $\Theta$ would need to be learned jointly and the parameter $\rho$ could be determined via cross validation on a held-out network. Extending the results in this paper to GLASSO will be the focus of subsequent research.

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
