[Reviews · NeurIPS 2014]

Submitted by Assigned_Reviewer_1

The authors learn a convolution filter for preprocessing signals from calcium imaging, which led them to be similarly effective at predicting neural connectivity from calcium imaging data as a winning algorithm in a recent competition on this problem. The winning algorithm used a grid search approach to preprocessing, but the approach described here will likely be more robust, and thus is of particular interest to a subset of the NIPS community.

1) Please plot f, n, and y on figure 1c in a more easy-to-visualize manner, as the relation between these is the crucial result supplied by this paper. Currently f and y are plotted but they are difficult to see due to the thick line-width and overlap.

Minor:
page 6 typo “fluoresce” should be fluorescence
Summary: This paper presents a useful approach to preprocessing calcium imaging data based on learning a convolution filter, which will likely be of interest to members of the NIPS community that work with such data.

Submitted by Assigned_Reviewer_43

The paper describes a method for estimating sparse connectivity graphs of firing neurons. An L_2 norm is used to obtain an penalised inverse covariance matrix as it improves the cost function. Furthermore, a previously published hard thresholding is replaced by a soft one. A cost function is formulated as the X-entropy and optimised using BFGS.

The paper is self-contained and the references are extensive. The text is very easy to read albeit the structure of the paper at times emerges from the text rather than being explicitly presented to the reader. The result is that the paper reads as if it is chronological and incremental rather than theoretically and scientifically motivated.

The paper compares favourably to other published methods, namely those who took part in the Kaggle Connectnomics competition. However, the heavily repeated reference to the competition makes the paper read like a late submission which, with the benefit of hindsight, is the best. This style also adds to the incremental feel of the publication.

Comments following author rebuttal:
I have decided to increase my score. What convinced me the the case of improved computational speed ("parametrized in a differentiable way with a very simple, easy-to-implement formula") and a better justification for the choices made, e.g. filter order and chi2 value.
Summary: The paper is well written and theoretically complete. Nevertheless, it reads like an belated contribution to the Kaggle competition showing an (albeit not insignificant) improvement using mainly logistic regression for preprocessing and an X-entropy formulation of the cost that allows the use efficient off-the-shelf solvers.

Submitted by Assigned_Reviewer_44

This paper addresses the reconstruction of network topology from calcium imaging data, using inverse covariance matrix estimation. It is shown empirically that a simple convolution filter (to be applied to the calcium traces) can be learned (once and for all) that substantially improves the reconstruction performance, and the time it takes to infer connectivity on new datasets.

----

QUALITY

Overall, what is done seems solid. There are a couple of wrinkles that I would like the authors to clarify.

First: the authors restricted the length of the convolution filter to 10 time steps. The learned filter peaks at time step 8, which left me to wonder whether there was enough room for the filter to "converge" (in "time lag space"). For example, had it been restricted to 5 time steps, the authors would have missed this presumably crucial peak. Was that a computational restriction? If not, is there a biophysical (or empirical?) reason why correlations would fall off anyway after 10 time steps? The authors should probably discuss this.

Second: although I understand that the study is focused on improving the inverse covariance method specifically, I would have liked to hear more of a discussion regarding the inherent limitations of the method. For example, the fact that only an undirected graph can be extracted seems like a big restriction as far as neural circuits are concerned.

Third, how crucial is it to get the value of $\chi$ in Eq.7 in the right ballpark? In particular, how does reconstruction generalize to datasets of very different sparsity than the one assumed for training?

CLARITY

The paper is well-written and well structured.

It'd be great if the Kaggle dataset (e.g. the fact that the ground truth connectivity is known) could be described upfront (at the beginning of the methods) in 1/2 sentences.

I would also like the authors to clarify the timescales for the non-experts (e.g. what does "one time step" in the convolution filter mean? what does "time (20 ms)" mean exactly in the x-axis labels? that 20ms are shown in total, or 1000*20ms = 20 sec?). It would also help the readers to appreciate the difficulty of the task.

Also, AUC is nowhere defined (not even spelled out -- Area Under the (ROC) Curve?).

ORIGINALITY

I'm not an expert in this specific calcium imaging literature, but it seems surprising that nobody had tried (even heuristic) convolution filters on top of calcium data prior to covariance estimation before... Anyhow, the work presented here is original, and clearly improves on the current leaderboard for Kaggle.

SIGNIFICANCE

Estimating network topology has very important implications for the neurosciences, especially with the advent of whole-network imaging techniques. There definitely is a need for statistical methods. While this study provides state-of-the-art performance and speed, I believe it remains essentially a simple (though important) addition to a known algorithm (L2-regularized inverse covariance estimation) which unfortunately does not address its most inherent limitations (e.g. underlying, implicit Gaussian assumption, undirected graph recovery, ...).
Summary: This paper is technically good, well written, and achieves state-of-the-art performance in (a specific case of) topology extraction from calcium data, an important problem in neuroscience. My greatest concern is the lack of a proper understanding for why the convolution filter learned here improves performance, and why the same filter can be re-used on other datasets with good generalization performance. It also looks like a rather incremental addition to a known algorithm.
Author Feedback
Author rebuttal: Comment: The heavily repeated reference to the competition makes the paper read like a late submission which, with the benefit of hindsight, is the best. This style also adds to the incremental feel of the publication.

Response: Inverse covariance estimation has been an active field of research in recent years (including several articles in NIPS), however to our knowledge no published articles develop methods for preprocessing of the data prior to applying inverse covariance. To quantify this contribution, direct application of inverse covariance achieves an AUC score of approximately 0.88 on the networks we consider, not significantly better than many other methods such as transfer entropy, discretized correlation, or granger causality. A dramatic improvement in accuracy is only achieved by using supervised signal processing for inverse covariance estimation (AUC approximately 0.94). We believe this result alone is worthy of publication.

While we are taking advantage of “competition hindsight”, we would argue that a research competition is less a definitive conclusion on a particular research topic, but instead a way to spark new research directions and to widen the research community in an area. In particular, neither the other solutions in the competition nor the methods developed in prior research articles made the leap that the inverse covariance operation can be parametrized in a differentiable way with a very simple, easy-to-implement formula. This allows inverse covariance to be coupled with a convolution step forming a “ConvOp” that can then be attached to a loss function in supervised learning. In real applications to inferring neural connectivity, we believe our method will be much more attractive to researchers than a method that takes weeks to train on a cluster, even if it was only discovered after the conclusion of the competition.

Comment: the authors restricted the length of the convolution filter to 10 time steps. The learned filter peaks at time step 8, which left me to wonder whether there was enough room for the filter to "converge" (in "time lag space"). For example, had it been restricted to 5 time steps, the authors would have missed this presumably crucial peak. Was that a computational restriction? If not, is there a biophysical (or empirical?) reason why correlations would fall off anyway after 10 time steps?

Response: There are two ways in which we determined the size of the convolution filter. The first is through inspecting the decay of cross-correlation as a function of the time-lag. For the networks we consider in the paper, this decay takes place over 10-15 time units. The second method is to add an additional time unit one at a time until cross-validated AUC scores no longer improve. This happens for the networks we consider at 10 time units. Presumably for other networks one may need bigger or smaller filters.

Comment: Although I understand that the study is focused on improving the inverse covariance method specifically, I would have liked to hear more of a discussion regarding the inherent limitations of the method. For example, the fact that only an undirected graph can be extracted seems like a big restriction as far as neural circuits are concerned.

Response: Empirically, using directed network estimation did not improve our methods, nor the methods of other top solutions. This may be due to the fact that the resolution of Calcium fluorescence imaging is coarser than the timescale of network dynamics, so that directionality information is lost in the imaging process.

That being said, it is possible to adapt our method for estimation of directed networks. This can be accomplished by introducing two different filters $\alpha_i$ and $\alpha_j$ into Equations 5 and 6 to allow for an asymmetric covariance matrix $S$ in Equation 6. It would be interesting to assess the performance of such a method on networks with higher resolution imaging in future research.

Comment: How crucial is it to get the value of $\chi$ in Eq.7 in the right ballpark? In particular, how does reconstruction generalize to datasets of very different sparsity than the one assumed for training?

Response: For $\chi$ values within 10% of the true value, AUC scores are above .935. Without data balancing, the model achieves an AUC score of .925, so the introduction of $\chi$ is important.

Comment: While this study provides state-of-the-art performance and speed, I believe it remains essentially a simple (though important) addition to a known algorithm (L2-regularized inverse covariance estimation) which unfortunately does not address its most inherent limitations (e.g. underlying, implicit Gaussian assumption, undirected graph recovery, ...).

Response: See response to comment 3 regarding directed graph recovery. Concerning the Gaussian assumption, if the input data $f_t^i$ was Gaussian and independent in $t$, then no signal processing would be necessary prior to applying inverse covariance estimation. However, neural firing time series data is generated by a nonlinear, mutually-exciting point process. We believe that it is the fact that the input data is non-Gaussian that the signal processing is so crucial. In this case $f_t^i$ and $f_s^j$ are highly dependent for $10>t-s>0$ and $j\rightarrow i$. Empirically, the learned convolution filter compensates for the model mis-specification and allows for the “wrong” model to still achieve a high degree of accuracy.